# Plaque Radiotherapy for Ocular Melanoma

**DOI:** 10.3390/cancers16193386

**Published:** 2024-10-03

**Authors:** George Naveen Thomas, I-Ling Chou, Lingam Gopal

**Affiliations:** 1Department of Ophthalmology, National University Health System, Singapore 119228, Singapore; 2Department of Ophthalmology, National University of Singapore, Singapore 119222, Singapore; 3School of Medicine, China Medical University, Taichung 404, Taiwan

**Keywords:** uvea, choroid, ciliary body, melanoma

## Abstract

**Simple Summary:**

This review serves as a comprehensive summary of how plaque radiotherapy, a specialized treatment for various eye tumors including uveal melanoma, is used to provide effective tumor control while preserving vision. The review discusses the radioisotopes available, explores various plaque designs, and describes clinical outcomes alongside possible complications. The review summarizes ongoing research and advancements that aim to enhance the effectiveness and safety of plaque brachytherapy, which is an important tool for the ocular oncologist.

**Abstract:**

Plaque radiotherapy is an effective treatment modality for medium-sized ocular tumors such as uveal melanoma. The authors review the available literature and concisely summarize the current state of the art of ophthalmic plaque brachytherapy. The choice of radioisotope, which includes Ruthenium-106 and Iodine-125, depends on the intended treatment duration, tumor characteristics, and side effect profiles. Ophthalmic plaques may be customized to allow for the delivery of a precise radiation dose by adjusting seed placement and plaque shape to minimize collateral tissue radiation. High dose rate (HDR) brachytherapy, using beta (e.g., Yttrium-90) and photon-emitting sources (e.g., Ytterbium-169, Selenium-75), allows for rapid radiation dose delivery, which typically lasts minutes, compared to multiple days with low-dose plaque brachytherapy. The efficacy of Ruthenium-106 brachytherapy for uveal melanoma varies widely, with reported local control rates between 59.0% and 98.0%. Factors influencing outcomes include tumor size, thickness, anatomical location, and radiation dose at the tumor apex, with larger and thicker tumors potentially exhibiting poorer response and a higher rate of complications. Plaque brachytherapy is effective for selected tumors, particularly uveal melanoma, providing comparable survival rates to enucleation for medium-sized tumors. The complications of plaque brachytherapy are well described, and many of these are treatable.

## 1. Introduction

It has been almost 100 years since R. Foster Moore described the first report of the insertion of radon seeds into the substance of what was described as a “melanotic sarcoma” [1]. Since then, there have been many publications describing the use of many other radioisotopes to treat intraocular tumors, including Cobalt-60, Ruthenium-106, Iodine-125, Palladium-103, Strontium-90 and Caesium-131 [2,3,4,5,6,7,8,9,10,11]. The Collaborative Ocular Melanoma Study (COMS) was the first clinical trial in choroidal melanoma that described standardized methodology for tumor diagnosis, radioactive plaque design, and treatment dosimetry [12,13]. Plaque radiotherapy, also known as ophthalmic brachytherapy, is now a well-established treatment for many ocular tumors. It is indicated in intraocular tumors that include uveal melanoma, choroidal hemangioma, retinoblastoma, vasoproliferative tumor, metastasis [14,15], and even ocular surface tumors such as ocular surface squamous neoplasia and conjunctival melanoma. Uveal melanoma is the most significant indication of these, being the most common primary intraocular malignancy in adults. Uveal melanoma is a malignant melanocytic cellular proliferation that is most common in the choroid (90%), the vascular layer located under the retina, but may also be located in the ciliary body (6%) or iris (4%) [16]. Its incidence varies with geography, with an incidence of 5.1 cases per million per year [17], 1.3–8.6 per million per year in Europe [18], and much lower in Asia and Africa at 0.2–0.3 per million per year. It is more common in Caucasian populations as compared to African American, Hispanic, and Asian patients [14]. Choroidal melanoma is most commonly seen as a dome-shaped mass inside the eye, which may be pigmented or non-pigmented [14]. It may remain asymptomatic until it is large enough to involve the visual axis or until there is fluid or blood that is caused by the tumor involving the visual axis [14]. The armamentarium of treatment modalities available to the ocular oncologist includes radiotherapy, chemotherapy, local resection, focal therapies such as laser thermal-based therapy or cryotherapy, and in advanced cases, enucleation. Types of radiation treatment include brachytherapy, also known as plaque radiotherapy, and external beam radiotherapy, which is available in many different forms including proton beam therapy and stereotactic radiotherapy [14].

## 2. Indications for Treatment

Tumors in various parts of the eye can be treated with plaque radiotherapy, including the choroid, retina [19], ciliary body [20], iris [21], and even the conjunctiva [22]. Types of tumors that can be treated include uveal melanoma [17], retinoblastoma [17], choroidal hemangioma [23], and even conjunctival melanoma [24], among many others.

As compared to enucleation, plaque radiotherapy is a preferred treatment option in ocular tumors, as it is a globe-sparing procedure. However, whether it is a first-line treatment remains controversial, and the best treatment plan depends significantly on individual tumor characteristics (e.g., size, location) [14].

Plaque radiotherapy is most frequently used to treat choroidal melanoma [25]. Medium-sized choroidal melanoma (defined as 2.5–10 mm apical height and ≤16 mm diameter) and small-sized melanoma exhibiting malignant behavior are the most common indications, according to the definition of the Collaborative Ocular Melanoma Study (COMS) [26].

Currently, with the advancement of plaque brachytherapy, its application has expanded past the original indication described in the COMS studies. The current American Brachytherapy Society (ABS) guidelines for plaque brachytherapy of uveal melanoma state that American Joint Commission on Cancer (AJCC) T1, T2, T3, and T4a–d uveal melanoma can be treated with brachytherapy; tumors with extraocular extension (T4e), basal diameters that exceed the limits of brachytherapy, painful blind eyes [27,28], and those with no light perception vision are not suitable for plaque therapy [19]. An analysis by Messer et al. from the National Cancer Database revealed that patient survival is improved in brachytherapy versus enucleation in all size cohorts, including large-sized choroidal melanoma [15,29,30,31].

It is the authors’ opinion that plaque brachytherapy is most suitable for medium to large tumors that are within the limits of the treatment modality available to the treating surgeon. Typically, tumors will range from 8 mm to 18 mm in base diameter, with a maximum thickness of 5–6 mm for Ruthenium-106 plaques and 10 mm for Iodine-125 plaques, with some variation based on surgeon experience.

## 3. Comparison with Other Treatment Options

Uveal melanoma may be treated with various modalities as described previously. The decision on the mode of treatment depends on the size and extent of the tumor within the eye. In general, small tumors can be treated with local therapies including laser and cryotherapy. Medium-sized tumors may be treated with either plaque brachytherapy, or external beam radiotherapy such as proton beam therapy, or enucleation. Large tumors have classically been treated with enucleation, although selected large tumors are treated in the current era with plaque radiotherapy, as compared to 10 or more years previously, when enucleation may have been performed instead in a similar case. There are no head-to-head clinical trials that have been published to date to the authors’ knowledge that have compared the various forms of treatment within each tumor size category. It is the authors’ observation that the main alternatives to plaque brachytherapy are proton beam therapy, stereotactic external beam radiotherapy, and enucleation. Plaque brachytherapy, proton beam therapy, and stereotactic radiotherapy are not universally available in all regions. In settings without the necessary expertise, enucleation is a safe and effective treatment. Various comparative analyses have found no significant differences in various morbidity and mortality-based outcomes between various forms of radiotherapy including plaque brachytherapy and external beam radiotherapy [32,33]. However, many advanced ocular oncology centers have been observed by the authors to employ either plaque brachytherapy or proton beam therapy in the treatment of appropriately selected cases of uveal melanoma.

## 4. Types of Radioisotopes and Comparison

Various radioisotopes, sizes and geometries of beta-radiation-emitting ophthalmic applicators are available for different tumor locations [34]. Although many radioisotopes have been employed for plaque radiation, including low-energy photon seeds (Iodine-125 [35], Palladium-103 [31], Cesium-131 [31], Cobalt-60 [36,37,38]), solid beta seeds [19] (Ruthenium-106 [3], and Strontium-90 [39,40]), and high dose seeds (Strontium-90 [41], Yttrium-90 [42], Ytterbium-169 and Selenium-75 [43], and Iridium-192 [44]), the two main radioisotopes in common use internationally are Ruthenium-106 and Iodine-125 [45].

Among the available radioisotopes, there are variations in radiation dose distribution, treatment efficacy, and side effect profile. Ruthenium-106 has been compared to Iodine-125 in terms of maximal possible tumor diameter and thickness, ocular complications, and treatment outcomes (enucleation, local recurrence, survival rate). Ruthenium-106 plaques have a longer half-life (371.8 vs. 59.4 days [46]) and therefore can be used for up to 1 year, while Iodine-125 seeds need to be replaced more frequently. Iodine-125 plaques have the advantage of being able to treat thicker tumors than Ruthenium-106 plaques [47]. Filì et al. demonstrated that thicker choroidal melanomas (≥5.5 mm) treated with Ruthenium-106 plaques had a significantly higher rate of repeated brachytherapy than with Iodine-125 seeds; however, rates of enucleation and patient survival were not significantly different [48]. Takiar et al. reported that at similar doses, Ruthenium-106 plaques had lower rates of ocular complications including radiation retinopathy and cataracts [49].

In North America, Iodine-125 and Palladium-103 are commonly selected for the treatment of choroidal melanoma with their large repositories and consistent supply chains. Like Iodine-125, treatment with Palladium-103 was associated with higher metastasis-free survival for thicker choroidal melanomas, compared to Ruthenium-106. On the contrary, there is some evidence that Ruthenium-106 may be associated with better visual acuity (VA) preservation and less radiation retinopathy [50,51]. In a systematic review of nine retrospective and three prospective studies [21], Palladium-103 showed a lower rate of glaucoma, improved effective dose, and lower mortality rate than Iodine-125. Novel applications of Palladium-103 on retinoblastoma and multifocal iris melanoma have also been investigated in small patient populations, demonstrating favorable local control as well as visual acuity and ocular and life preservation [52,53].

Given the limited evidence regarding differences between radioisotopes in terms of efficacy, complication rate, and survival outcome, there is no definitively superior radioisotope. Currently, it is the author’s observation that the selection of isotope type is largely influenced by their accessibility and surgeon preference, with Iodine-125 being more popular in North America, and Ruthenium-106 in Europe. A summary of common radioisotopes used in plaque brachytherapy is given in Table 1.

## 5. Plaque Size, Materials, and Dosimetry Innovations

Photon-emitting plaques come in various designs. The photon-emitting seeds can be affixed to the surface of the plaque or placed within an insert; the distance of the seeds from the tumor, the heights of the lip and the plaque, and the curvature of the plaque can be customized [34]. The most commonly used plaques are COMS-style plaques, consisting of a gold-alloy backing, and a Silastic seed carrier insert, with sizes ranging from 10 mm to 22 mm in diameter containing 5 to 24 seeds [54,55,56].

There have been several modifications to the COMS-style plaques for specific indications. Plaques may be round or notched, the latter of which allows treatment of tumors adjacent to the optic nerve, which fits into the notch, and allows an optimized dose delivery to these tumors [57].

Various techniques have been described to create tumor models or simulations, with the aim of more precise treatment planning [56,58,59]. Creating a precise model of the tumor, considering its irregular shape, is expected to yield a treatment plan that conforms more closely to the tumor contours, consequently leading to a potentially improved treatment outcome and reduction in the dose received to other normal tissues [60]. For example, Studenski et al. employed plaque simulation software capable of precisely accommodating irregularities in the tumor base. Instead of a simple elliptical base, this software optimizes tumor coverage while minimizing radiation exposure to normal ocular structures by modeling the actual tumor contour [60]. The authors noted significant improvements in dosimetry, especially if the tumor was located <4 mm from the fovea and <2 mm from the optic disk.

Alterations in plaque materials could potentially decrease treatment side effects and facilitate the process of insertion. The British Columbia Cancer Agency designed an acrylic plaque with Gold-198 seeds that showed minimal toxicity with adequate globe preservation, survival, and local control rate in 79 patients [61]. Plaques with stainless steel backing such as the Australian (ROPES) eye plaque demonstrated a lower dose rate outside of the eye [62]. Astrahan et al. designed a reusable gold seed-guide insert that allows for easier and faster positioning of seeds [63]. Although many modifications have been described, it is the authors’ observation that standard plaque designs available through established international suppliers, when used correctly, allow for a very high success rate of local control in appropriately selected cases of uveal melanoma.

## 6. Surgical Technique

Prior to treatment planning, the assessment and measurement of the tumor characteristics are performed to allow the surgeon to decide on a suitable plaque size, material, and intended radiation dosage. These measurements could be obtained using a full dilated ophthalmic examination, multimodal imaging including color fundus photography, autofluorescence imaging, B-scan ultrasonography, and optical coherence tomography imaging. Tumor basal diameter and apical height measurements will influence the size of the plaque. Additionally, the plaque diameter should be slightly larger than the tumor to allow a margin of safety. Typically, the plaque is sized with a diameter of 2 mm greater than the diameter of the tumor base. Plaque planning softwares, such as the Plaque Simulator 6 by Eye Physics [64], can also assist in the selection of plaque type, placement, and radiation dosage calculation.

The plaque application surgery could be performed either under general or regional anesthesia in an operating room. The tumor size and location can be rechecked with indirect ophthalmoscopy immediately prior to the commencement of surgery. A 180° to 270° peritomy is performed to approach the Tenon’s capsule. Ocular muscles including rectus and oblique muscles are isolated and retracted with silk sutures to make room for plaque positioning. The muscles are disinserted if necessary as demonstrated in Figure 1A,B. The tumor location is verified with transpupillary or transocular illumination 180° away from the tumor; its base will shadow its subjacent sclera. A sterile surgical marking pen is used to outline the edges of the shadow on the sclera. Then, the plaque is placed over the marked area. It is suggested to mark the border of the plaque to further ensure an additional 2 mm of “margin” around the tumor. Intraoperative ultrasound could be performed to ensure proper positioning of the plaque. The plaque is then sutured over the marked area as seen in Figure 1C,D. Sutures could also be pre-placed with a plaque template or a dummy plaque, and the template is then replaced with the actual plaque once its position is confirmed, to reduce the potential exposure of radiation to the operating surgeon. Next, the location of the plaque is verified with transillumination, indirect ophthalmoscopy, or ultrasound. In the authors’ experience, in pigmented eyes, tumor localization with indirect ophthalmoscopy may be more successful than transillumination, as the latter may be difficult because of associated choroidal pigmentation, which makes the tumor shadow more difficult to see in these eyes. Ultrasonographic confirmation of tumor position may lead to more accurate positioning of the plaque [65,66]. Once the plaque is inserted as shown in Figure 1E,F, the tumor is assessed for bleeding and the optic nerve for adequate perfusion. The extraocular muscles are then repositioned or reattached as presented in Figure 1G,H. One possible option for handling an extraocular muscle that needs to be detached is to tag the muscle with a 5-0 black silk suture and let it remain detached under the conjunctiva. At the time of removal of the plaque, the muscle can be reattached, and the tagging suture removed. This avoids unnecessary handling of the muscle during the removal of the plaque. Finally, the eye is irrigated with an antibiotic solution and the conjunctival peritomy is closed with sutures.

Throughout the treatment period, the eye is patched and covered. A lead shield may be used to cover the eye to minimize leakage of radiation outside the orbit. Once the treatment is completed, the plaque is removed under regional or general anesthesia [19,67].

## 7. High Dose Rate (HDR) Brachytherapy

Presently, low-dose-rate (LDR) brachytherapy predominates ophthalmic brachytherapy. However, high-dose-rate (HDR) brachytherapy is undergoing active research and development for both beta-particle and photon-emitting options [68].

High-dose-rate (HDR) brachytherapy is a form of radiation therapy where a radioactive source is temporarily placed in or near the tumor, delivering a high dose of radiation in a short period of time. Unlike low-dose-rate (LDR) brachytherapy, where the radiation is delivered continuously over a longer period, HDR brachytherapy delivers the radiation in brief sessions typically lasting a few minutes. This technique allows for precise targeting of the tumor while minimizing radiation exposure to surrounding healthy tissues [69]. Only one surgery is needed, as the radioactive plaque is inserted and removed during one surgery, unlike two surgeries in conventional low-dose brachytherapy.

Beta radiation has long been utilized in the treatment of ophthalmic tumors and benign growths, with high-dose-rate Strontium-90 or Yttrium-90 being a common and extensively cited approach for preventing pterygium-related episcleral fibrovascular growths [41]. Finger and colleagues have published multiple studies of their experience with Yttrium-90 [68,70,71]. In a recent case report [72], high-dose-rate Yttrium-90 brachytherapy was employed to treat a cT1a-category choroidal melanoma. They used a “Liberty Vision Yttrium-90 disc–iWand P device” with a disk activity of 15.48 mCi and a diameter of 6.0 mm. The episcleral application lasted for 3 min and 39 s, resulting in a high-dose rate of 30 Gy at the tumor’s apex, located 1.6 mm from the episclera. At the 13-month follow-up, reductions in subretinal fluid (SRF) and tumor thickness were observed. Furthermore, there were no occurrences of secondary cataract, radiation retinopathy, maculopathy, or optic neuropathy, and the patient achieved a visual acuity of 20/20. The disk of this module was limited to prescription doses ranging from 22 to 30 Gy, depths of 1.6–2.6 mm, and treatment durations of 219–773 s [66]. Despite these current limitations, beta-radiation HDR brachytherapy offers a novel option for single-surgery, minimally invasive, outpatient irradiation [73].

Photon-emitting particles are also utilized for HDR brachytherapy. Dupere et al. experimented with Ytterbium-169 [42] and Selenium-75 [43] in two respective studies of middle-energy HDR brachytherapy sources. They designed a gold shielded applicator with radiation sources within, with a ring-like configuration, collimating the beam. The “MCNP^®^” software package simulation was used to calculate the dosimetry. Both sources were tested on various tumors ranging from 3.5 to 8 mm apex depth and 10 to 15 mm in diameter. The results demonstrated that middle-energy HDR brachytherapy could decrease the maximum absorbed dose to critical structures including posterior lens, iris, optic nerve, and sclera, compared to traditional isotopes such as Iodine-125 and Ruthenium-106 [70,71]. It is the observation of the authors that high-dose brachytherapy is not in widespread clinical use yet, but this may change as new developments are made.

## 8. Adjunctive Procedures: Cytology and Cytogenetics

Currently, primary treatment for uveal melanoma (UM) includes radiation therapy, resection, and enucleation [74]. Despite good initial rates of tumor regression, advances in management have not significantly improved mortality rates. Around 50% of patients develop metastatic disease, with a typical survival of 6 to 12 months [75]. To address the large variability in the prognosis of patients, various biomarkers may be used to determine which patients are more at risk of metastasis and mortality [76,77,78,79,80,81].

Importantly, chromosomal abnormalities in tumor tissue have been linked with treatment outcomes. Damato et al. collected data of 356 patients alongside their chromosome 3 and chromosome 8 genetic data using fluorescence in situ hybridization (FISH). The findings revealed that chromosomes 3 and 8 exhibited improved predictive power for treatment outcomes, with basal tumor diameter, monosomy 3, and epithelioid cellularity being the most significant predictive factor of disease-specific mortality [82]. Metastatic uveal melanoma has also been correlated with BAP1 and BRCA1 gene mutations [83,84], though the exact pathogenesis remains unclear [85].

The liver is the predominant site for metastasis, through the hematogenous route [86,87,88,89,90]. New blood-serum-based biomarkers are under evaluation to predict and monitor treatment response, including immune markers [73,91,92], growth factors [42,93], beta-2-microglobulin molecules [43,74,94], microRNAs [80,95], and circulating tumor cells [75,96].

In the authors’ experience, cytogenetics of tumor tissue as obtained via needle biopsy in the operating theater is useful both to the patient and the treating physician to provide an indication of the local recurrence and metastatic risk. On the other hand, cytology may be required only if the diagnosis is uncertain, as uveal melanoma can usually be diagnosed clinically.

## 9. Efficacy of Treatment

The efficacy of plaque radiotherapy that is reported in the scientific literature is varied. Regarding the efficacy of Ruthenium-106 brachytherapy for uveal melanoma, the local control rate ranges from 59.0% [97] to 98.0% [98,99]. Karimi et al. performed a meta-analysis of 21 studies including 3913 affected eyes; it showed an overall efficacy of 84% in the local control of uveal melanoma with Ruthenium-106 brachytherapy [100].

Due to the limited depth penetration of radioisotopes, the thickness and size of uveal melanoma might serve as a predictive factor for the efficacy of brachytherapy, with thicker and larger tumors potentially exhibiting reduced treatment response and more radiation complications compared to smaller tumors [101]. Furthermore, previous studies have revealed that a thicker tumor depth and larger basal tumor diameter was significantly predictive of metastasis [102,103].

The anatomical location of the tumor and the radiation dosage delivered to its apex appear to be supplementary factors that influence the therapeutic response to brachytherapy. For uveal melanoma, the local control rate is lower in certain anatomical locations such as the ciliary body [104], juxtapapillary region [105], or the foveal center [97]. Visual outcomes deteriorate when tumors are closer to visually critical structures like the optic disk or foveola [80]. Radiation dosage was also found to be a crucial determinant of local control in brachytherapy. There was a positive correlation between local control rates and the mean dose at the apex [106]. It is the authors’ experience that plaque brachytherapy is a safe and effective treatment with local control exceeding 95% in well-selected cases. The efficacy may be reduced in cases, which are approaching the maximum treatable tumor dimensions or when positioning of the plaque is technically challenging for various anatomical reasons, such as a smaller orbit, higher intra-orbital pressure, or reduced palpebral aperture.

## 10. Complications of Plaque Radiotherapy and Its Treatment

A summary of complications of plaque radiotherapy and its treatment with their respective mechanism and management is listed in Table 2.

### 10.1. Scleral and Corneal Complications

Radiation exposure to the corneal epithelium, conjunctival goblet cells, and even limbal stem cells may lead to keratitis and dry eye [108], which was reported in 3.8–8.3% of cases within a period of 20.7 to 34.0 months. For tumors situated near the lacrimal gland, radiation may cause atrophy and result in a type of dry eye syndrome called keratoconjunctivitis sicca [114]. However, most tumors are located in the posterior segment and, therefore, these symptoms may occur as a result of alterations to the conjunctiva rather than direct radiation-induced damage to the lacrimal gland. Consequently, this may be much more significant for patients with conjunctival melanoma treated with plaque radiotherapy. Dry eye symptoms can be treated with topical lubricants and gels.

### 10.2. Neovascular Glaucoma

Upregulation of vascular endothelial growth factor (VEGF), as well as damage to the iris vasculature, may lead to proliferation of new vessels on the iris, also known as *rubeosis iridis*. If the angle becomes involved, it may lead to the development of neovascular glaucoma (NVG), which can rapidly lead to severe visual loss. In fact, NVG stands as the most common cause for secondary enucleation [113]. With Iodine-125 brachytherapy, this is reported to occur in 4–23% of treated eyes at a mean of 26.7 months [115]. The main risk factors linked with NVG include advanced age, larger tumor size, anterior tumor location, and elevated baseline intraocular pressure (IOP) [116]. Increased risk of rubeosis has been seen with disinsertion of a horizontal rectus muscle, possibly due to anterior segment ischemia [113]. Larger tumors have also been associated with rubeosis [106]. NVG carries a poor prognosis and can necessitate enucleation, underscoring the importance of early diagnosis and treatment. Neovascular glaucoma can be treated with established medical and surgical treatments that include topical intraocular pressure lowering medication, oral acetazolamide, laser cyclophotocoagulation, and even glaucoma filtration surgery.

### 10.3. Radiation-Induced Cataract

Lens fibers can be damaged by ionizing radiation. The eye exposed to radiation may develop a posterior subcapsular cataract, cortical cataract, or nuclear sclerotic cataract, with posterior subcapsular cataracts being the most common. It could present as vacuoles, scattered granules, and even as a mature white cataract with exposure to higher radiation doses [109]. The COMS found that the rate of vision-limiting cataract formation was 68% in 5 years, and was the primary reason for diminished vision after treatment [117]. This large variation may depend on the distance of the radiation seeds to the lens and the dose of radiation prescribed to the tumor [118]. With a cumulative dose of 24 Gy or more to the lens, the incidence of cataract was 92%, whereas those with 12 Gy or less had an incidence of 12% [113]. Radiation-induced cataract is treated with similar techniques compared to the more common age-related cataract. The COMS group reported the outcomes of cataract surgery in patients with Iodine-125 plaques after 5 years and found that median visual acuity (VA) improved from 20/125 to 20/40, and the rate of improvement by 2 lines on the Snellen VA chart was 66% [113]. Furthermore, cataract surgery for these patients poses no greater risk of complications compared to cataracts not associated with ionizing radiation [21].

### 10.4. Radiation-Induced Retinopathy and Maculopathy

Ionizing radiation may affect the retina and macula. Closure of the vascular bed is a well-known effect of ionizing radiation [119]. This may result in intraretinal hemorrhage, cotton-wool spots, non-perfusion of retinal vessels, neovascularization of the retina, cystoid macular edema, and, in its later stages, vitreous hemorrhage [120]. The incidence of radiation retinopathy and maculopathy depends on the dose of radiation exposed to the retina and macula, with the risk being increased for more posterior tumors and consequently lower for tumors involving the more anterior structures such as the ciliary body [110]. The incidence of radiation retinopathy and maculopathy was reported to be between 10 and 63% and 13 and 52% [115] and the mean duration from plaque brachytherapy treatment to the onset of radiation retinopathy was found to be approximately 25 months [113]. Numerous treatment modalities have been described in the treatment of radiation retinopathy and maculopathy, including intravitreal injection of anti-vascular endothelial growth factor (anti-VEGF), intravitreal injection of steroid such as triamcinolone, and laser photocoagulation. Intravitreal anti-VEGF and steroid injections are typically used for treating macular edema, while laser photocoagulation may be applied in the area of the ischemic retina to treat radiation retinopathy [113]. Shields and colleagues have reported good results with intravitreal bevacizumab administered every 4 months for 2 years, as compared to controls, with less cystoid macular edema on OCT (44% vs. 54%; *p* = 0.01), radiation papillopathy (6% vs. 12%, *p* = 0.04), and better Logarithm of the Minimum Angle of Resolution (LOGMAR) visual acuity at 48 months (0.54 [20/70] vs. 2.00 [counting fingers], *p* = 0.04) [121].

### 10.5. Radiation-Induced Optic Neuropathy

Plaque radiotherapy administered to tumors in close proximity to the optic nerve would allow a higher dose of radiation to the optic nerve, which could possibly result in radiation-induced optic neuropathy. Signs of these in the eye include optic disk hyperemia, edema, circumpapillary hemorrhage, and cotton-wool spots. Other factors that would make this more likely include higher radiation dose to the optic disk and larger tumors [111,115,122]. This has been reported in 8–16% of cases treated with Iodine-125 plaque brachytherapy [21], at an interval of between 16 and 22 months post-brachytherapy [112,118]. Most cases of radiation-induced optic neuropathy result in severe vision loss. Shields and colleagues reported that 7 out of 9 patients receiving injections of intravitreal triamcinolone acetate (4 mg in 0.1 mL) had visual acuity stability or improvement, with all patients having resolution of optic disk hyperemia and edema at 11 months post-injection [123]. Hyperbaric oxygen therapy has been used in attempts to treat radiation retinopathy, but good results have only been demonstrated in a few patients [124].

### 10.6. Visual Acuity

In the Collaborative Ocular Melanoma Study (COMS) report number 16, 49% of patients had a visual acuity loss of 6 or more Snellen lines [27] after Iodine-125 plaque brachytherapy. In the COMS, the factors of ‘tumor apical height > 5 mm’ and ‘distance between tumor and foveal center < 2 mm’ were the strongest predictors of a loss of 6 or more Snellen lines of visual acuity. Other factors that were also associated with reduced visual acuity included coexistent diabetes, tumor-associated retinal detachment, and non-dome-shaped tumors [122]. Other studies have found that radiation dose, greater tumor size, and close proximity of the optic nerve head and the macula were associated with visual acuity loss [4,112,125,126,127,128,129]. Jones and colleagues reported that a radiation dose rate of 111 cGy/h to the macula predicted a 50% risk of visual acuity loss [4].

### 10.7. Extraocular Muscle Dysfunction and Diplopia

It has been reported that approximately 60% of patients exhibited ocular alignment abnormalities in one case series [130]. However, only 10% of patients in the same series reported diplopia [112]. Dawson and colleagues found that only 1.7% of patients out of 929 patients reported diplopia across a period of 8 years [131]. Wen and colleagues reported their personal experience that a significant proportion of patients develop transient diplopia after Iodine-125 plaque brachytherapy, but that most of the cases spontaneously resolve after the first month [115]. There is no well-established treatment protocol for the management of diplopia after plaque brachytherapy. Cases that do not resolve after the first month may be considered for prisms and botox injections in the first 6 months, after which definite strabismus surgery should only be performed when the degree of ocular deviation has been stable for more than 6 months [112].

## 11. Conclusions

Plaque brachytherapy is widely considered to be an effective treatment for selected cases of ocular tumors, the most significant being uveal melanoma. In essence, it offers excellent local control of focal tumors and serves as a viable substitute for enucleation of medium-sized tumors, with comparable survival rates. Recent innovations in plaque design have optimized radiation distribution within the eye, reducing the likelihood of local treatment failure. With many refinements in technique over the last two decades, plaque brachytherapy remains an essential tool in the arsenal of the ocular oncologist.

## Figures and Tables

**Figure 1 cancers-16-03386-f001:**
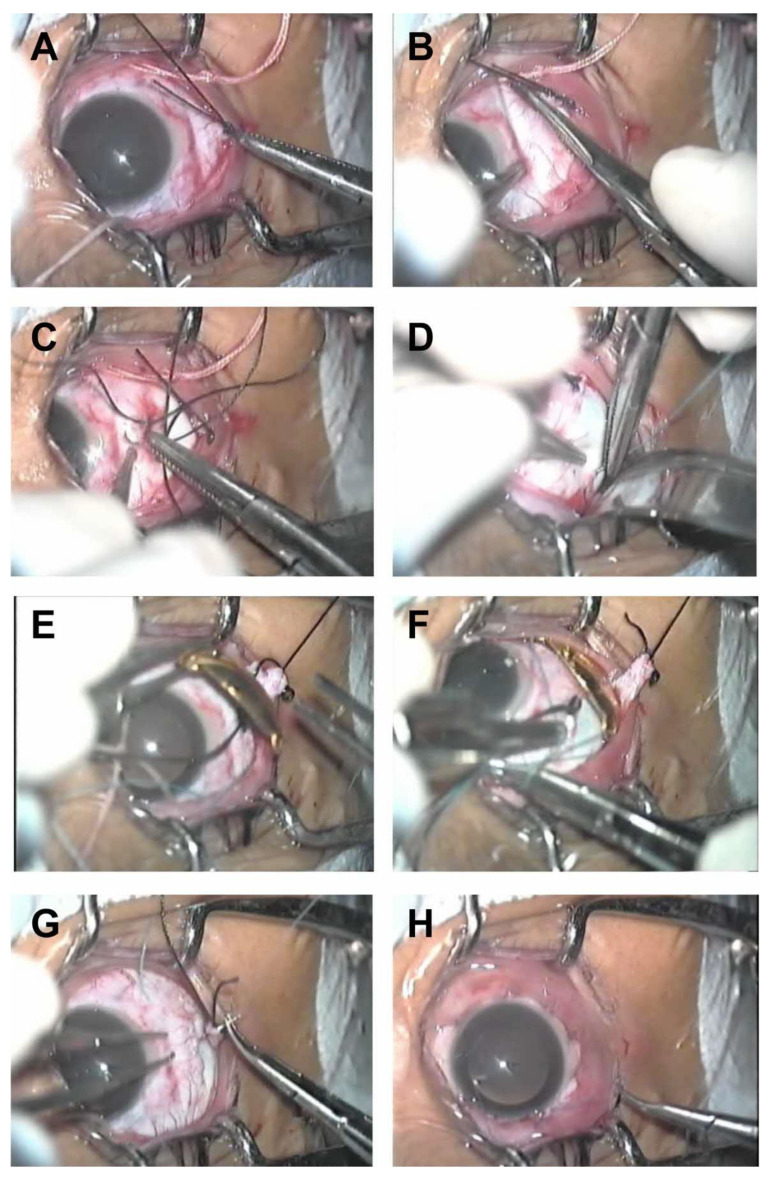
Plaque insertion surgery. (**A**) Tagging the lateral rectus muscle; (**B**) disinserting the lateral rectus muscle; (**C**) passing a traction suture through the stump of the insertion of lateral rectus muscle; (**D**) passing a scleral suture at the intended location of the anterior eyelet of plaque; (**E**) placement of the gold plaque with I-125 seeds; (**F**) anchoring the plaque to the sclera; (**G**) reattaching the detached lateral rectus; (**H**) conjunctival closure.

**Table 1 cancers-16-03386-t001:** Summary of common radioisotopes used in plaque brachytherapy.

General information	Most commonly used isotopes include Iodine-125 [31], Ruthenium-106 [34], Palladium-103 [18], Strontium-90 [37], and Cobalt-60 [32]. No clear superior isotope based on current evidence in terms of efficacy, complication rates, and survival outcomes. Selection is often influenced by accessibility and surgeon preference.
Ruthenium-106 [34]	Longer half-life (371.8 days) allows for use for up to 1 year; Iodine-125 needs more frequent replacement [41]. Possibly lower rates of ocular complications such as radiation retinopathy and cataracts compared to Iodine-125 [44]. Associated with potential better visual acuity preservation and less radiation retinopathy [45]. Thicker choroidal melanomas (≥5.5 mm) treated may require repeated brachytherapy more frequently [42].
Iodine-125 [31]	Can treat thicker tumors compared to Ruthenium-106 plaques [42]. Commonly used in North America alongside Palladium-103 for choroidal melanoma due to increased availability. Higher metastasis-free survival reported for thicker choroidal melanomas compared to Ruthenium-106 [45].
Palladium-103 [18]	Possibly lower rates of glaucoma reported compared to Iodine-125 [47]. Improved effective dose and lower mortality rate in systematic reviews [47]. Investigated novel applications for retinoblastoma and multifocal iris melanoma with favorable outcomes in small studies [48].

**Table 2 cancers-16-03386-t002:** Summary of complications of plaque radiotherapy and its treatment.

Complication	Mechanism	Management
Keratitis and dry eye syndrome	Radiation exposure during plaque radiotherapy may lead to goblet cell and limbal stem cell injury [101].	Topical lubricants and gels
Neovascular glaucoma	Upregulation of vascular endothelial growth factor (VEGF) and iris vasculature damage can cause rubeosis iridis, potentially progressing to neovascular glaucoma (NVG), which carries a poor prognosis and may require enucleation if untreated [107].	Topical eye-pressure-lowering medications, oral acetazolamide, laser cyclophotocoagulation, glaucoma filtration surgery [107]
Radiation-induced cataract	Ionizing radiation can damage lens fibers, leading to posterior subcapsular, cortical, or nuclear sclerotic cataracts, with posterior subcapsular cataracts being most common [108].	Cataract surgery, treated similarly to age-related cataracts [108]
Radiation-induced retinopathy and maculopathy	Ionizing radiation can lead to closure of the retinal vascular bed, causing intraretinal hemorrhage, cotton-wool spots, non-perfusion of vessels, neovascularization, cystoid macular edema, and vitreous hemorrhage [109].	Macular edema: intravitreal anti-VEGF and steroid injections Ischemic radiation retinopathy: laser photocoagulation, anti-VEGF injections [109]
Radiation-induced optic neuropathy	Characterized by optic disk hyperemia, edema, hemorrhage, and cotton-wool spots [110].	Intravitreal triamcinolone acetate injections, hyperbaric oxygen therapy [110]
Visual acuity loss	Various; linked to higher radiation doses, larger tumor size, and proximity of the tumor to critical structures [111].	Treatment of the specific cause
Extraocular muscle dysfunction and diplopia	Intraoperative muscle disinsertion, radiation-induced extraocular muscle dysfunction [112].	May recover spontaneously; prisms and botulinum toxin injections for symptoms persisting <6 months; strabismus surgery if symptoms persist >6 months [113]

## Data Availability

No new data were created or analyzed in this study. Data sharing is not applicable to this article.

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
