# Peer review of "Plaque Radiotherapy for Ocular Melanoma"

_cancers, 2024, doi:10.3390/cancers16193386_

Round 1

Reviewer 1 Report

Comments and Suggestions for Authors

Please, give more details about history - background, when the certain technique was invented etc.

Please, give more details about comparison of brachytherapy with other irradiation techniques, gamma knife, linear accelerator, proton beam irradiation duie to localizations of the tumor and volume etc...

Please, check the References - not correctly cited e.g. 21, 108

In References is missing the classification scheme for using several treatment options and comparison with other irradiation techniques

Author Response

We thank the reviewers for their insightful comments and suggestions. We have carefully considered these and have revised our manuscript, and we feel that it is now a significantly stronger contribution.

Comment 1: Please, give more details about history - background, when the certain technique was invented etc.

Thank you for the insightful comment. We have taken this excellent suggestion on board and have described in Line 1 Page 1 onwards the first description of the use of radon seeds for an intraocular tumour in 1930 by Moore and the history of the development of the various radioisotopes.

Comment 2: Please, give more details about comparison of brachytherapy with other irradiation techniques, gamma knife, linear accelerator, proton beam irradiation due to localizations of the tumor and volume etc...

This is an excellent suggestion and we have included a new section on this, from Line 91 onwards describing the other forms of treatment and also the other forms of radiation treatment types.

Comment 3: Please, check the References - not correctly cited e.g. 21, 108

Thank you again for the feedback. We have checked the references and paid particular attention to those references. Here is a summary of the statement and associated reference:

"Although many radioisotopes have been employed for plaque radiation including low-energy photon seeds (Iodine-125, Palladium-103, Cesium-131, Cobalt-60[21])"

Reference 21: Fass, D.; McCormick, B.; Abramson, D.; Ellsworth, R. Cobalt-60 plaques in recurrent retinoblastoma. Int. J. Radiat. Oncol. Biol. Phys. 1991, 21, 625–627.

"Shields and colleagues have reported good results with intravitreal bevacizumab ad-ministered every 4 months for 2 years, as compared to controls, with less cystoid macu-lar oedema on OCT (44% vs. 54%; p = 0.01), radiation papillopathy (6% vs. 12%, p = 0.04) and better Logarithm of the Minimum Angle of Resolution (LOGMAR) visual acu-ity at 48 months (0.54 [20/70] vs. 2.00 [counting fingers], p = 0.04)"

Reference 108: Shields, C.L.; Dalvin, L.A.; Chang, M.; Mazloumi, M.; Fortin, P.; McGarrey, M.; Martin, A.; Yaghy, A.; Yang, X.; Vichitvejpaisal, P.; Mashayekhi, A.; Shields, J.A. Visual outcome at 4 years following plaque radiotherapy and prophylactic intravitreal bevacizumab (every 4 months for 2 years) for uveal melanoma. JAMA Ophthalmol. 2020, 138, 136–146.

Comment 4: In References is missing the classification scheme for using several treatment options and comparison with other irradiation techniques

This is an a useful suggestion and we have included in conjunction with the new section developed that discusses the various other forms of therapy, from Line 91 onwards describing the other forms of treatment and also the other forms of radiation treatment types.

We thank the Reviewer for their insightful comments and we feel that Reviewer's comments have strengthened our comprehensive review article.

Reviewer 2 Report

Comments and Suggestions for Authors

Summary: A scoping review on the plaque radiotherapy for ocular melanoma.

Comments: 

1. Introduction_line 55: The term "linear accelerator therapy" is not a common term in Radiation oncology. "linear accelerator" is the device which produce the external beam radiotherapy. It is suggested another term is used.

2. line 70: authors are suggested to mention the numerical values of the clinical studies, in addition to stating the "significant number of eyes". This information can reveal the extent of plaque RT benefit.

3. line 84–85: kindly add the iridium 192 to the list.

4. line 105–108 "However, Powell et al. demonstrated  that in high-risk choroidal melanoma patients, early administration of intravitreal anti-VEGF medication (bevacizumab) after plaque therapy could prevent vision-threatening 107 radiation maculopathy. " This study has loose connection with the previous and next sentences.

5. Tables 1 and 2. Kindly add the references for each given information.  

6. It's suggested to add a figure outlining the section "Plaque size, materials and dosimetry innovations ". 

7. It's suggested to add "intra-operative surgery" and "graphical" images to the section "Surgical technique". 

8. line 222: "Adjunctive procedures: Cytology and cytogenetics ". The importance/role of cytogenetic analysis in opting the adjunctive treatments has not been fully explained. This paragraph is somehow confusing for me as a reader. It is suggested to revise this section to adhere its heading. 

9. It's suggested the authors add a section comparing the clinical studies comparing plaque radiotherapy versus other radiotherapy modalities. 

10. The importance of intra-operative sonography in localization of ocular tumor during plaque radiotherapy needs to be highlighted [by referring the clinical studies]. 

11. It's suggested the authors add their own opinions (as experts in the field) on the optimal suggested treatment planning, including the choice on radioisotope, dosage, planning. These pieces of information can be added to the end of the corresponding sections. 

Author Response

We thank the reviewers for their insightful comments and suggestions. We have carefully considered these and have revised our manuscript, and we feel that it is now a significantly stronger contribution.

Reply to comments: 

1. Introduction_line 55: The term "linear accelerator therapy" is not a common term in Radiation oncology. "linear accelerator" is the device which produce the external beam radiotherapy. It is suggested another term is used.

Thank you, this point is well taken and has been updated in our manuscript.

2. line 70: authors are suggested to mention the numerical values of the clinical studies, in addition to stating the "significant number of eyes". This information can reveal the extent of plaque RT benefit.

We agree with the Reviewer that this statement was imprecise and irrelevant to the topic. We have removed that line to increase clarity in that section. We have included a comment in the manuscript to reflect this change.

3. line 84–85: kindly add the iridium 192 to the list.

This is a good observation and we have included reports that have included Iridium-192.

4. line 105–108 "However, Powell et al. demonstrated  that in high-risk choroidal melanoma patients, early administration of intravitreal anti-VEGF medication (bevacizumab) after plaque therapy could prevent vision-threatening 107 radiation maculopathy. " This study has loose connection with the previous and next sentences.

Again, we agree with the Reviewer that this statement was irrelevant to the topic. We have again removed that line to increase clarity in that section. We have included a comment in the manuscript to reflect this change.

5. Tables 1 and 2. Kindly add the references for each given information.  

This is an excellent suggestion and we have included references for both tables.

6. It's suggested to add a figure outlining the section "Plaque size, materials and dosimetry innovations ". 

We thank the Reviewer for the suggestion. We wanted to highlight uncommon deviations in standard practice for the sake of completeness since this was designed to be a comprehensive review. The vast majority of centres using plaque brachytherapy use standardised techniques and these innovations are not in common practice. However, we did not want to emphasise them too much, in order to avoid giving the reader the impression that these are commonly employed or even familiar to most ocular oncologists. We have suggested leaving out this section for now so as not to confuse the reader.

7. It's suggested to add "intra-operative surgery" and "graphical" images to the section "Surgical technique". 

This is an excellent suggestion. We have included Figure 1, which contains 8 surgical pictures showing the critical steps of plaque brachytherapy surgery.

8. line 222: "Adjunctive procedures: Cytology and cytogenetics ". The importance/role of cytogenetic analysis in opting the adjunctive treatments has not been fully explained. This paragraph is somehow confusing for me as a reader. It is suggested to revise this section to adhere its heading. 

We agree with the Reviewer that this section needed to be significantly clearer and clinically relevant. It has been extensively revised, with changes tracked.

9. It's suggested the authors add a section comparing the clinical studies comparing plaque radiotherapy versus other radiotherapy modalities. 

We agree that this would be a beneficial addition. We have included an entirely new section to address this point, in Line 91 onwards.

10. The importance of intra-operative sonography in localization of ocular tumor during plaque radiotherapy needs to be highlighted [by referring the clinical studies]. 

The Reviewer is obviously familiar with this technique which is an excellent suggestion, and we have included in Line 210 a comment and 2 references that emphasise how ultrasound may improve the position of the plaque and improve outcomes.

11. It's suggested the authors add their own opinions (as experts in the field) on the optimal suggested treatment planning, including the choice on radioisotope, dosage, planning. These pieces of information can be added to the end of the corresponding sections. 

We appreciate this comment and have added the opinions and experiences of the authors in treating these tumours and have reflected these to be the personal experiences of the authors. We feel that this has added clinical relevance to the manuscript.

We would like to extend special thanks to the Reviewer for their insightful and helpful comments.

Round 2

Reviewer 1 Report

Comments and Suggestions for Authors

x

Author Response

We thank the reviewer for their insightful comments and suggestions on new references that could be added. We have taken the opportunity to incorporate these into the manuscript and feel that it strengthens the paper.

On the comment of the surgical technique, the description is comprehensive and represents the correct technique that is applicable for plaque insertion, regardless of radioisotope.

Additionally, we have included a comprehensive discussion on the complications of plaque brachytherapy and its associated treatment, under clear headings. If there are specific concerns, we would be most happy to address these.

We would like to take this opportunity to thank the reviewer and the editorial team for your comments and suggestions so far.
